# Reproducibility Report: D3S - A Discriminative Single Shot Segmentation Tracker

1 ## Reproducibility Summary

2 **Scope of Reproducibility**

3 The original paper describes the architecture of the D3S neural network and evaluates its performance in the task of
4 visual object tracking and video segmentation tasks. In our reproducibility study, we focused on training and evaluation
5 of D3S for visual object tracking tasks due to limited time.

6 **Methodology**

7 Our work is based on code provided by the authors of the original paper. The training code was reorganized and partially
8 re-implemented. As a result, our version consists of only the most necessary code (the original code consists of other
9 experiments not presented in the paper). For model evaluation, we use the pytracking framework following the authors
10 of the original article. We used *NVIDIA Tesla V100* GPU with *CUDA 9.2* and *pytorch 1.7.1* for model training and
11 validation. The time it took to train the model was 16 hours.

12 **Results**

13 The difference of the reproduced model quality metrics does not exceed 3%. These differences do not change the
14 position of D3S relative to other architectures in comparison. It is found that the speed of model evaluation (FPS)
15 differs significantly for different datasets, whereas the original paper provided a single estimate of a speed. At the same
16 time, the obtained values are lower than the ones given in the article. The reason for the differences may be the various
17 hardware configurations of the computers used for the experiments.

18 **What was easy**

19 The open-source code of the authors was very helpful. Also, the evaluation pipeline in visual object training is not
20 trivial, and the authors of the original code use the pytracking framework for this task. It is significantly reduced the
21 complexity of our work.

22 **What was difficult**

23 We had a few problems due to incompatibilities between the versions of *pytorch* and *CUDA* used in the original code
24 and required to work with our hardware. In addition, it is not clear from the original paper how metrics were calculated
25 from the raw output (bounding boxes): by toolkits supplied with datasets or somehow else.

26 **Communication with original authors**

27 We did not communicate with the authors at all, except to use their publicly available source code.

| Metrics | D3S | SPM | SiamMask | ATOM | ASRCF | SiamRPN | CSRDCF | CCOT | TCNN |
|---------|-----|-----|----------|------|-------|---------|--------|------|------|
| EAO | **0.493** | 0.434 | 0.433 | 0.430 | 0.391 | 0.344 | 0.338 | 0.331 | 0.325 |
| Acc. | **0.66** | 0.62 | 0.64 | 0.61 | 0.56 | 0.56 | 0.51 | 0.54 | 0.55 |
| Rob. | **0.131** | 0.210 | 0.214 | 0.180 | 0.187 | 0.302 | 0.238 | 0.238 | 0.268 |

Table 1: VOT2016 – comparison with state-of-the-art trackers.

| Metrics | D3S | SiamRPN++ | ATOM | LADCF | DaSiamRPN | SiamMask | SPM | ASRCF |
|---------|-----|-----------|------|-------|-----------|----------|-----|-------|
| EAO | **0.489** | 0.414 | 0.401 | 0.389 | 0.383 | 0.380 | 0.338 | 0.328 |
| Acc. | **0.64** | 0.60 | 0.59 | 0.51 | 0.59 | 0.61 | 0.58 | 0.49 |
| Rob. | **0.150** | 0.234 | 0.204 | 0.159 | 0.276 | 0.276 | 0.300 | 0.234 |

Table 2: VOT2018 – comparison with state-of-the-art trackers.

# 1 Introduction

The most common formulation of visual object tracking considers the task of reporting target location in each frame of the video given a single training image. D3S - a discriminative single shot segmentation tracker [1] is a single shot network that applies two target models with complementary geometric properties, one invariant to a broad range of transformations, the other assuming a rigid object. D3S was trained on youtube-VOS 2018 dataset only for segmentation as the primary output and evaluated on vot2016, vot2018, GOT10-k, and TrackingNet datasets without per-dataset finetuning. D3S outperforms other state-of-the-art trackers on most of these tracking benchmarks.

# 2 Scope of reproducibility

The original paper demonstrates the results of D3S evaluation on visual object tracking and video object segmentation datasets, but in our reproducibility study, we focused on training and evaluation of D3S for visual object tracking tasks only due to limited time. The results of comparisons of D3S with other neural networks architectures on various benchmarks from the original papers are shown in tables 1 - 4. The main claims of the original paper are as follows:

- Claim 1: D3S outperforms state-of-the-art trackers on the VOT2016, VOT2018 and GOT-10k benchmarks and performs on par with top trackers on TrackingNet, regardless of the fact that some of the tested trackers were retrained for specific datasets.
- Claim 2: D3S evaluation speed close to real-time (25fps) on a single *NVidia GTX 1080* GPU.
- Claim 3: D3S significantly outperforms recent top segmentation tracker SiamMask on all benchmarks in all metrics and contributes towards narrowing the gap between two, currently separate, domains of short-term tracking and video object segmentation, thus blurring the boundary between the two.

# 3 Methodology

Our work is based on the code provided by the authors of the original paper. The code consists of two parts - training code and evaluation code based on the pytracking framework [2]. The training code consists of neural network

| Metrics | D3S | ATOM | SiamMask | SiamFCv2 | SiamFC | GOTURN | CCOT | MDNet |
|---------|-----|------|----------|----------|--------|--------|------|-------|
| AO | **59.7** | 55.6 | 51.4 | 37.4 | 34.8 | 34.2 | 32.5 | 29.9 |
| SR0.75 | **46.2** | 40.2 | 36.6 | 14.4 | 9.8 | 12.4 | 10.7 | 9.9 |
| SR0.5 | **67.6** | 63.5 | 58.7 | 40.4 | 35.3 | 37.5 | 32.8 | 30.3 |

Table 3: GOT-10k test set – comparison with state-of-the-art trackers.

| Metrics | D3S | SiamRPN++ | SiamMask | ATOM | MDNet | CFNet | SiamFC | ECO |
|---------|-----|-----------|----------|------|-------|-------|--------|-----|
| AUC | 72.8 | **73.3** | 72.5 | 70.3 | 60.6 | 57.8 | 57.1 | 55.4 |
| Prec. | **66.4** | 69.4 | 66.4 | 64.8 | 56.5 | 53.3 | 53.3 | 49.2 |
| Prec.N | **76.8** | 80.0 | 77.8 | 77.1 | 70.5 | 65.4 | 66.3 | 61.8 |

Table 4: TrackingNet test set – comparison with state-of-the-art trackers.

description, hyperparameter settings, training cycle, etc. We have revised and reorganized this part of the code to leave only the code necessary to investigate reproducibility.

## 3.1 Model descriptions

The backbone features are extracted from the target search region resized to $384 \times 384$ pixels. The backbone network in D3S is composed of the first four layers of ResNet50, pre-trained on ImageNet for object classification. Two models are used in D3S to robustly cope with target appearance changes and background discrimination: a geometrically invariant model (GIM) and a geometrically constrained Euclidean model (GEM). The GIM and GEM pathways provide complementary information about the pixel-level target presence. GEM provides a robust, but rather inaccurate estimate of the target region, whereas the output channels from GIM show greater detail, but are less discriminative. These models process the input in parallel pathways and produce several coarse target presence channels, which are fused into a detailed segmentation map by a refinement pathway. A refinement pathway is thus designed to combine the different information channels and upscale the solution into an accurate and detailed segmentation map. For a more detailed description see the original paper [1].

## 3.2 Datasets

The model was trained on the YouTube-VOS dataset [3] (2018 version, train part, 3471 sequences). The model evaluation was carried out on VOT2016 and VOT2018 [4] ( 60 sequences each), GOT10-k (180 test sequences), TrackingNet [5] (511 test sequences) datasets.

## 3.3 Hyperparameters

We used hyperparameters settings provided in the original paper [1]: batch size 64, 40 epochs training with 1000 iterations per epoch, ADAM optimizer with a learning rate set to 0.001 and with 0.2 decay every 15 epochs.

## 3.4 Experimental setup and code

Firstly, datasets were downloaded (by URL or using toolkits) and the model was trained. Model evaluation was carried out by the pytracking framework [2], which generates output files with target bounding boxes for each frame of each sequence. For metric calculation we used toolkits supplied with datasets. For exact commands see code and description in supplementary materials (Readme.md).

## 3.5 Computational requirements

Our work was performed using resources of the NRNU MEPhI high-performance computing center. For model training end evaluation we used *NVIDIA Tesla V100* GPU with *CUDA 9.2* and *pytorch 1.7.1*. The training time was 16 hours. Evaluation speed will be reported in section 4.

## 4 Results

For evaluated D3s on the visual object tracking benchmarks and calculated metrics to support claims 1 and 2 of the original paper. A comparison of the results obtained with those given in the original article is shown in table 5. Evaluation speeds (FPS) listed in table 6.

| Dataset | Metric | Our result | Original result |
|---|---|---|---|
| vot2016 | EAO | 0.494 | 0.493 |
| | Acc. | 0.67 | 0.66 |
| | Rob. | 0.131 | 0.131 |
| vot 2018 | EAO | 0.487 | 0.489 |
| | Acc. | 0.63 | 0.64 |
| | Rob. | 0.153 | 0.150 |
| GOT10-k | AO | 0.60 | 59.7 |
| | $SR_{0.75}$ | 47.3 | 46.2 |
| | $SR_{0.5}$ | 68.6 | 67.6 |
| TrackingNet | AUC | 72.8 | 72.8 |
| | Prec. | 66.5 | 66.4 |
| | $Prec._N$ | 76.8 | 76.8 |

Table 5: Comparison of the results

| Dataset | vot2016 | vot2018 | GOT10-k | TrackingNet |
|---|---|---|---|---|
| Our result | 22 | 21 | 16 | 23 |
| Original result | | | 25 | |

Table 6: Comparison of the results

# 5 Discussion

The difference of the reproduced model quality metrics does not exceed 3%. These differences do not change the position of D3S relative to other architectures in comparison. It is found that the speed of model evaluation (FPS) differs significantly for different datasets, whereas the original paper provided a single estimate of a speed. The resulting speeds of the model are lower than those indicated in the original article but still close to real-time. The reason for the differences may be the various hardware configurations of the computers used for the experiments. Thus, the data sets in our case were stored on a separate machine connected to the computational node with the GPU over the network. This could have affected the speed degradation.

## 5.1 What was easy

The open-source code of the authors was very helpful. Also, the evaluation pipeline in visual object training is not trivial, and the authors of the original code use the pytracking [2] framework for this task. It is significantly reduced the complexity of our work.

## 5.2 What was difficult

We had a few problems due to incompatibilities between the versions of *pytorch* and *CUDA* used in the original code and required to work with our hardware. In addition, it is not clear from the original paper how metrics were calculated from the raw output (bounding boxes): by toolkits supplied with datasets or somehow else.

## 5.3 Communication with original authors

We did not communicate with the authors at all, except to use their publicly available source code.

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
