# OpenReview forum: "Reproducibility Report: D3S - A Discriminative Single Shot Segmentation Tracker"
_ML_Reproducibility_Challenge/2021/Fall — Reject_

### Official Review · Reviewer_mkTu · 2022-02-23
**Review on Reproducibility Report: D3S - A Discriminative Single Shot Segmentation Tracker**

**Rating:** 4
**Confidence:** 5

**Review:**

The paper is clear and well written. The authors have tried to reproduce a specific part of the previous work (D3S - A Discriminative Single Shot Segmentation Tracker).
The authors have removed the unnecessary part of the code from the original paper and trained and evaluated them on the dataset for visual object tracking.

The authors report noticeable changes in the speed of model evaluation and at the same time mention that their data have been stored on a separate machine than the computing machine. There are not enough details provided on the file system, network traffic, etc during the model evaluation. But it is possible that the data being stored on a separate machine has caused the issue, which suggests that the I/O has slowed down the whole process.

Pros:
Authors have been able to reproduce the original results with at most 3% difference.

Cons:
I do not really see anything original or of significance about this work. The hyperparameter setting is the same as the original paper. There are no ablation studies, recommendations for improvement, new hyperparameter search, or any added novelty beyond the original work. As a result, I do not recommend this work for acceptance.

---

### Official Review · Reviewer_z6Bq · 2022-03-01
**Close reproduction with some minor errors in writing**

**Rating:** 6
**Confidence:** 3

**Review:**

The reproduction differs from the original results by 3%. The experimental procedure described looks like a very good match to what's in the original paper.

The reproducers also made code changes from the original code. They were not able to run the original code due to environment setup issues with pythorch and CUDA. From what I can tell from the submission, it is uncertain whether the source of the 3% difference is from a) the original papers results not being precisely reproducible or b) the reproducers' code changes. It would be good to dig into where the discrepancy comes from in more detail, especially with more empirical results.

Though the reproduction matches the original results by a margin of much less than 3% for most results.

What are the units in Table 5? Is the "our result" for GOT-10k really that much lower? Or is it just an inconsistency on whether the numbers are in %? Plase clarify. Consistency with the original paper would be ideal, but the original paper does switch units as well between Tables 1&2 and 3&4. It's also confusing that the main paper switches between EAO vs AO for unclear reasons (and the reproduction does too). It'd be helpful to clarify what the metrics are and in what units they are presented in the tables.

Minor comment: Grammar in 1st sentence of section 4 is confusing.

Overall, I'd say this is a reasonable reproduction with a fairly close match. It could be executed and described more exactly and completely, though.

---

### Meta-Review · Program_Chairs · 2022-04-07

**Recommendation:** Reject
**Confidence:** 4

**Metareview:**

While the reproduction of the original effort has been done, there are no useful insights that have been produced by the authors of the current study, which is a necessary part of the challenge. See comments from reviewer mkTu for more details that can be added to the paper to make it stronger.

---

### Decision · Program_Chairs · 2022-04-09

Reject